# Interpretable GWAS by linking clinical phenotypes to quantifiable immune repertoire components
Yuhao Tan [1,2,3], Lida Wang [4], Hongyi Zhang [2,3], Mingyao Pan [2,3], Dajiang J. Liu [4,6] ✉,
Xiaowei Zhan [5,6] ✉ & Bo Li [1,2,3,6] ✉

Bridging the gap between genotype and phenotype in GWAS studies is challenging. A multitude of genetic variants have been associated with immune-related diseases, including cancer, yet the interpretability of most variants remains low. Here, we investigate the quantitative components in the T cell receptor (TCR) repertoire, the frequency of clusters of TCR sequences predicted to have common antigen specificity, to interpret the genetic associations of diverse human diseases. We first developed a statistical model to predict the TCR components using variants in the TRB and HLA loci. Applying this model to over 300,000 individuals in the UK Biobank data, we identified 2309 associations between TCR abundances and various immune diseases. TCR clusters predicted to be pathogenic for autoimmune diseases were significantly enriched for predicted autoantigen-specificity. Moreover, four TCR clusters were associated with better outcomes in distinct cancers, where conventional GWAS cannot identify any significant locus. Collectively, our results highlight the integral role of adaptive immune responses in explaining the associations between genotype and phenotype.

Genome-wide association studies (GWAS) have successfully identified numerous genetic variants linked to autoimmune diseases and cancers[1,2]. However, the mechanisms underlying these associations remain largely obscure[3]. Identification of disease-associated genes and cellular processes have partially explained disease etiology[4–7], and immune phenotypes, such as immune cell proportion and cytokine production, also provided mechanistic insights into some diseases[8,9]. However, direct associations between the T cell receptor (TCR) repertoire, a critical component in the immune system, remain insufficiently explored. The TCR is pivotal in adaptive immune response[10,11], as it recognizes antigen epitopes bounded by the human leukocyte antigen (HLA) molecule, which is encoded by the major histocompatibility complex (MHC) locus in Chromosome 6. T cells participate in a wide spectrum of human diseases, including autoimmune diseases, cancers, and infectious diseases[12]. Therefore, understanding the genetic influence of TCR repertoire will provide insights into the disease susceptibility, etiology and clinical outcome in the general population. Genetic variants influencing TCR sequence generation have been identified[13–16], setting the stage to investigate the impact of these genotype-TCR associations on disease predisposition.

A recent study provided genetic evidence supporting the hypothesis that HLA risk alleles shape T cell repertoire during thymic selection[15]. However, their association analysis was limited to three autoimmune diseases (celiac disease (CD), type 1 diabetes (T1D), and rheumatoid arthritis (RA)), without covering the effects in other diseases. Besides, they examined only known autoimmune risk variants, whereas TCR repertoires could be influenced by other variants. Additionally, the study used amino acid compositions of the CDR3 repertoire for quantification, which are not directly related to antigen specificity and T cell phenotypes. To address these limitations, we used the recently developed embedding method, Repertoire Functional Units (RFUs)[17], for TCR repertoire quantification. This method divides the TCR space into a fixed number of RFUs, each representing a cluster of TCRs with similar sequences. To quantify a given TCR repertoire, we assigned each TCR sequence to an RFU by identifying the nearest neighbor in the embedding space, and used the abundance of each cluster as a numerical encoding of the repertoire (Supplementary Fig. 1).

In this study, we provide more direct evidence to support the role of TCR repertoires in variant-disease associations leveraging UK Biobank (UKBB)[18], a large cohort with extensive clinical information. We developed

[1]Graduate Group in Genomics and Computational Biology, University of Pennsylvania Perelman School of Medicine, Philadelphia, PA, 19104, USA. [2]Center for Computational and Genomic Medicine, The Children's Hospital of Philadelphia, Philadelphia, PA, USA. [3]Department of Pathology and Laboratory Medicine, University of Pennsylvania, Philadelphia, PA, USA. [4]Institute for Personalized Medicine, College of Medicine, Pennsylvania State University, Hershey, PA, USA. [5]Quantitative Biomedical Research Center, Peter O'Donnell School of Public Health, University of Texas Southwestern Medical Center, Dallas, TX, USA. [6]These authors jointly supervised this work: Dajiang J. Liu, Xiaowei Zhan, Bo Li. ✉e-mail: dajiang.liu@psu.edu; xiaowei.zhan@utsouthwestern.edu; lib3@chop.edu

a lasso-based model to predict RFU abundances based on genetic variants. Applying this model to UKBB samples, we predicted RFU abundances and discovered their associations with various diseases, including autoimmune, hematopoietic, and metabolic disorders. Besides, we identified variant sets associated with cancer prognosis that could not be detected by conventional GWAS. Our novel analytical framework provides a systematic approach to understand the impact of TCR repertoires across the phenome, presenting them as immunologically meaningful variant aggregates to clarify the complex links between genotypes and a broad spectrum of diseases.

## Results

### Genome-wide association of TCR repertoire units

We analyzed a cohort comprising both whole genome sequencing (WGS) and longitudinal peripheral blood RNA sequencing (RNA-seq) data (rfuQTL training set; Supplementary Table 1)[19]. Utilizing TRUST4[20], we identified TCR sequences from RNA-seq data, minimizing the amplification bias inherent in TCR sequencing (TCR-seq) data[21]. We quantified RFU abundances by counting the number of TCR sequences in each TCR repertoire unit. No significant correlation between RFU abundances and the sample age were observed when assessed at distinct time points (Spearman's $|\rho| \geq 0.2$, FDR < 0.05; Supplementary Fig. 2a). Hence, we aggregated TCR sequences over multiple time points of the same individual to get sufficient TCR sequences for RFU abundance calculations (Supplementary Fig. 2b). After quality control (Methods, Supplementary Fig. 2c, d), the dataset included 659 individuals, with an average of 11,304 unique CDR3 beta chains per individual (Fig. 1a, b).

We used linear regression to test the associations between whole-genome variants and RFU abundances ($n = 4953$) after quantile normalization and correction for sex and the first five genotype principal components (PCs; Methods, Supplementary Fig. 3a–c). This normalization was necessary to prevent suspicious associations caused by extreme values. Additionally, similar to eQTL analysis[22], we adjusted for the first two probabilistic estimation of expression residual (PEER)[23] factors to account for hidden confounders, including sequencing depth, to avoid spurious associations (Supplementary Fig. 3d, e). We found 2,083 significant associations ($P < 10^{-11}$; Fig. 1d, Supplementary Fig. 4a, Supplementary Data 1). Adjusting for cell type proportions did not affect the association results (Supplementary Fig. 4b). We identified 2076 associations in the T cell receptor beta (TRB) locus, which is crucial for V(D)J recombination in TCR sequence generation. The most significant association was a missense variant in the non-coding region of TRBV 12-5 associated with the abundance of RFU 1415 (Fig. 1c). Consistently, TCRs assigned to RFU 1415 contained the initial sequence of the TRBV 12-5 CDR3 region ("CASGL")[24], suggesting that the variants may regulate the expression of TRBV 12-5. Additionally, we observed a substantial number of rfuQTL variants in non-coding regions, including one variant in TRB enhancer and MTA2 binding site, linked with RFU 2811 (Supplementary Fig. 4c).

To further explain the roles of these variants, we analyzed the enrichment across different classes of regulatory elements within the ENCODE Registry candidate cis-Regulatory Elements (cCREs; $n = 5$). We found a significant enrichment of rfuQTL variants in proximal enhancer-like signature (pELS; $P = 1.3 \times 10^{-12}$, one-sided Fisher's Exact Test; Fig. 1e). Additionally, we tested the enrichment of rfuQTL variants in transcription factor binding sites using the ENCODE3 Transcription Factor ChIP-seq Peaks dataset ($n = 324$). We observed significant enrichments for MTA2 and GATA1 ($P < 1.5 \times 10^{-4}$; Fig. 1f), with MTA2 previously linked to T cell activation and V(D)J recombination in B cells[25,26]. These results collectively suggested that both enhancers and specific transcription factors within the TRB locus played a crucial role in shaping the TCR repertoire.

We identified seven associations in the HLA locus, known for its influence in shaping the TCR repertoire through thymic selection (Example in Supplementary Fig. 4d). To better interpret the functional consequence of the associated alleles, we also performed association testing between imputed HLA haplotypes ($n = 28$) and normalized RFU abundances. We identified 19 significant associations, including three in the MHC-I locus

and sixteen in the MHC-II locus (Supplementary Table 2). RFUs with stronger associations with MHC-II than MHC-I are expected to be enriched with CD4 T cells, given their role in recognizing antigens presented by MHC-II. To test this hypothesis, we annotated CD4 and CD8 RFUs using TCR sequencing (TCR-seq) data of sorted CD4 and CD8 T cells (Supplementary Table 3; Supplementary Fig. 5a, Supplementary Data 2)[27,28]. We observed a correlation between the CD4/CD8 abundance ratio in an RFU and the MHC-II/MHC-I effect size ratio (Spearman's $\rho = 0.12$, $P = 3.5 \times 10^{-3}$, Supplementary Fig. 4e), indicating that our RFUs were reflective of cell-type specificity.

To reproduce these results, we analyzed an independent adult cohort dataset with both HLA haplotype and TCR-seq data (628 individuals; rfuQTL test set, Supplementary Table 1)[29]. After applying similar quality controls and covariate adjustments (Methods, Supplementary Fig. 3f, g), we observed a consistent effect size of the association between HLA haplotype and RFU abundances (Pearson's $r = 0.50$, $P = 5.0 \times 10^{-10}$; Supplementary Fig. 4f). Together, these findings aligned with previous studies highlighting the impacts of genetic polymorphism of the TRB and HLA loci on V(D)J gene usage[13], and extended these observations to the relative abundance of TCR clusters in a repertoire.

### Genetic prediction of RFU levels with regularized regression model

We next sought to predict the RFU abundances using genetic variants since TCR repertoire data is usually not available in large cohorts. We called the predicted RFU abundances as genetically determined RFU (gdRFU). We employed both lasso and elastic net models to predict normalized and covariate-corrected RFU abundances based on several genetic variant sets, using variants from TRB, HLA, or TRB and HLA locus (TRB + HLA). We assessed the model performance through ten-fold cross-validation. The number of well-predicted RFUs of lasso model is larger than Elastic Net (Supplementary Table 4), and thus we focused on lasso model in the downstream analysis. Additionally, we utilized Genome-wide Complex Trait Analysis (GCTA)[30] to estimate the heritability of these traits, which is the variance that could be explained by variants in the chosen variant set, setting a benchmark for the upper limits of our prediction accuracy. We observed that using variants solely from the TRB locus or a combination of variants from both TRB and HLA loci yielded comparable predictive outcomes and heritability estimates (Supplementary Fig. 6a, 6b). Consequently, we selected the most effective model from these variant sets.

We then defined RFUs with a cross-validated $R^2$ greater than 0.01 as predictable RFUs ($\geq 10\%$ correlation between predicted and observed abundance), following common practice in transcriptome-wide association studies[31]. The prediction performance of our model was validated using the 398 individuals with imputed genotype data in rfuQTL test set. After pre-processing and prediction (Methods), we calculated the heritability of RFU abundances and the $R^2$ between predicted abundances and actual corrected abundances. Our model successfully identified 1,351 predictable RFUs, achieving a mean cross-validated $R^2$ of 0.025 and a mean predictive $R^2$ of 0.011 for these RFUs (Fig. 2a). We observed a strong correlation between model performance in cross-validation and the test set (Pearson's $r = 0.50$, $P = 6.2 \times 10^{-85}$, Fig. 2b), revealing the model's robustness and transferability, given the variations in genetic architecture, variant quality, sequencing methodologies, and participant ages across the two datasets. Two illustrative examples of these predictions were presented in Fig. 2d and e.

We next annotated the RFUs for T cell phenotypic subsets, including naïve cells (TN), central memory cells (CM), regulatory T cells (Treg), and stem cell-like memory T cells (Tscm) in a public TCR-seq dataset with sorted T cells (Supplementary Table 3)[32]. We calculated the fraction of each cell type in each RFU, and found predictable RFUs exhibited a significantly higher fraction of TN cells and a significantly lower fraction of Treg cells (Wilcoxon signed-rank test, $P = 1.3 \times 10^{-10}$ for TN and $P = 1.2 \times 10^{-9}$ for Treg; Fig. 2c, Supplementary Fig. 7). This observation is consistent with the previous observation that the TCR sequences of the naïve cells were

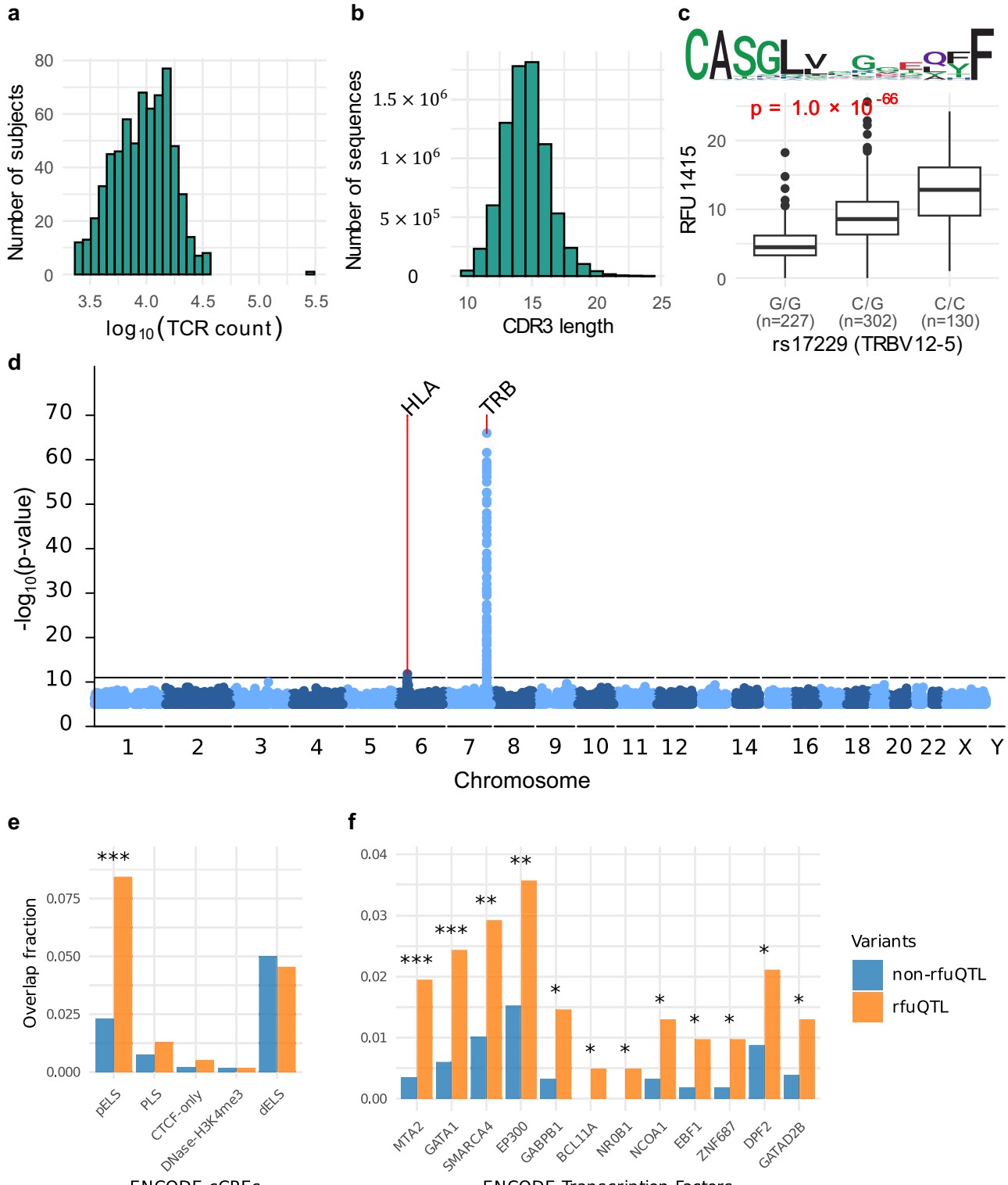

**Fig. 1 | Variants at TRB and HLA loci associate with RFU abundances. a** the number of unique CDR3 sequences per subject in rfuQTL training set. **b** the distribution of CDR3 amino acid length in the training set. **c**, an example of variants-RFU associations. The top panel displays the RFU motif, while the bottom part illustrates the unnormalized RFU abundances corresponding to different variants. **d** $P$ values of the association between whole genome variants and RFU abundances. Only associations with $P<10^{-5}$ were plotted. The black horizontal line indicates a Bonferroni-corrected $p$-value significance threshold of $10^{-11}$. **e** the proportion of both rfuQTL and non-rfuQTL variants that overlap with ENCODE cCREs, relative to their total numbers. pELS, proximal enhancer-like signature. PLS, promoter-like signature. dELS, distal enhancer-like signature. **f** the proportion of variants that overlap with ENCODE3 Transcription Factor ChIP-seq Peaks. Only transcription factors with $P$ value < 0.01 were plotted. Significance is evaluated using a one-sided Fisher's Exact Test, with * indicating $P < 0.01$, ** $P < 0.001$, and *** $P < 0.0001$. The boxplots represent the median (horizontal line within the box), the interquartile range (IQR; the box itself), and the whiskers, which extend to the maximum and minimum values lying within 1.5 times the IQR from the box. Data points outside the whiskers are individually plotted.

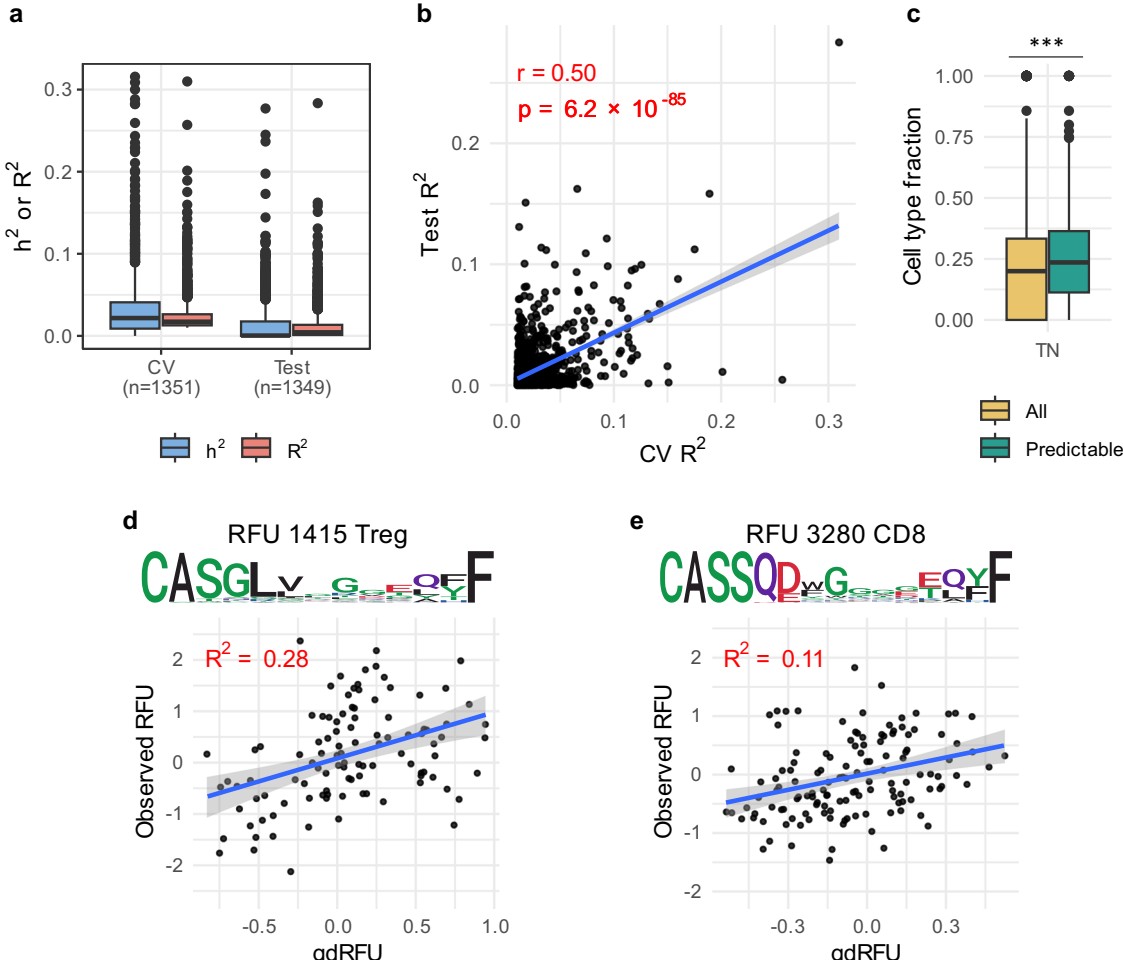

**Fig. 2 | Lasso models predict RFU abundances from genetic variants. a** the heritability ($h^2$) and performance ($R^2$) of cross validation (CV) on rfuQTL training set ($n = 659$) and testing on rfuQTL test set ($n = 398$). **b** the cross-validated and test performance. **c** the proportion of true naïve T cell within each RFU, comparing the distribution across all RFUs and predictable RFUs. Significance is evaluated using Wilcoxon signed-rank test, with *** indicating $P<0.0001$. **d, e** examples of RFU prediction. The RFU number, cell type, and motif are shown at the top, and a comparison of observed versus predicted values in rfuQTL test set is shown at the bottom.

more heritable compared to memory cells[33]. We then annotated RFUs as the cell type with the highest fraction if the abundances of different cell types are significant different (Friedman test, $P<3.7 \times 10^{-5}$). In total, we identified 3 Tscm RFUs, 9 CM RFUs, 61 TN RFUs, and 95 Treg RFUs within predictable RFUs (Supplementary Fig. 5b, Supplementary Data 3).

### gdRFU abundances associated with autoimmune diseases
We next performed a systematic RFU-wide association analysis (RfuWAS) utilizing individual-level data from the UKBB[18] (RfuWAS dataset; Supplementary Table 1). After quality control and gdRFU predictions (Methods), the dataset contained 1086 diseases (by phecodes) and 1351 gdRFU abundances of 337,122 unrelated individuals, after removing relatives up to the second degree. We performed linear regression between diseases and gdRFU abundances, using first ten genotype PCs, sex, and age as covariates, and identified 2309 significant associations ($P<3.4 \times 10^{-8}$; Fig. 3a, b, Supplementary Fig. 8a, Supplementary Data 4). These associations showed a marked enrichment in autoimmune diseases (Chi-squared test, $P<2.2 \times 10^{-16}$). Among the eight diseases with over 50 gdRFU associations, five were with autoimmune diseases, two were with thyroid diseases having potential autoimmune etiologies[34], and one involved iron metabolic disorder, where iron was related with T cell development[35]. By comparing our results with GWAS associations in the Pan UK Biobank[36], we identified that the kidney calculus (Phecode 594.1) was associated with RFU 2579 ($p = 7.1 \times 10^{-10}$) but no variants in the TRB and HLA loci were associated

with this condition. T cells have been implicated in the formation of kidney stones[37], and it is possible that selected T cell clones are related to the disease.

We then assessed the proportion of disease-associated gdRFUs in different cell types determined in the previous sections. We found a large proportion of central memory gdRFUs are disease-associated, with an enrichment for Type 1 Diabetes (T1D), CD, and two thyroid disorders (Fisher's exact test, $P<0.0125$; Fig. 3c). Additionally, we compared the proportion of disease-associated gdRFUs in CD4 and CD8 cell types. While more CD4 gdRFUs exhibited disease associations, this increase was not significant (Supplementary Fig. 8c). Restricted within pathogenic gdRFUs (gdRFUs positively associated with disease), the proportion of disease-associated RFUs became significant higher from random in diseases like T1D and thyroid disorders (Fisher's exact test, $P<0.025$, Supplementary Fig. 8d).

We next investigated the antigen-specificity of these pathogenic gdRFUs. As described in our previous benchmark, TCRs assigned to the same RFU have 90% of the chance to be specific to the same pool of antigens. To demonstrate that our previous benchmark results were generalizable to diverse antigens, we repeated the benchmark with ten randomly selected epitopes. We found that the RFU method achieved similar performance, revealing its robustness across different antigens (Supplementary Data 5, Supplementary Fig. 8b). Therefore, the specificity of an RFU can be evaluated with the TCRs of known antigen-specificity. We analyzed such TCRs specific to CD-related antigens (gliadin) and T1D-related antigens (mainly

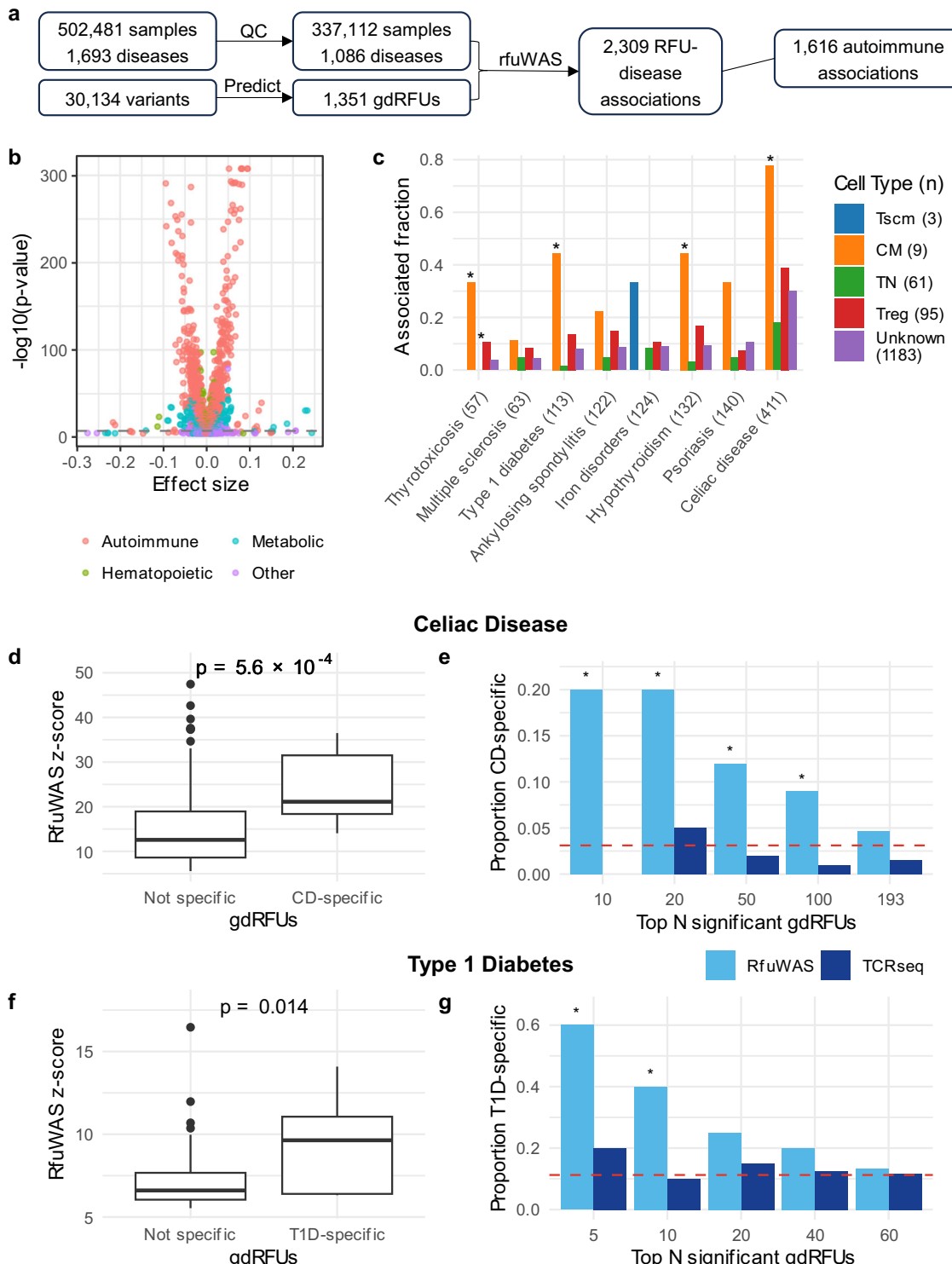

**Fig. 3 | Disease associations with gdRFU abundances. a** the workflow of RfuWAS. QC, quality control. **b** Volcano plot showing the associations between various diseases and RFU abundances. We include only associations with $P<3.7 \times 10^{-5}$. The dashed line represents the Bonferroni-adjusted $p$-value threshold. Associations are color-coded by disease types, with types representing fewer than 100 associations classified as "other". **c** the fraction of disease associated RFUs within each cell type. The numbers of associated RFUs are indicated in parentheses following each disease type, and the numbers of RFUs per cell type are indicated in parentheses following each cell type. **d, f** boxplots of z-scores of disease-specific versus non-specific RFUs in disease association analyses for CD (**d**) and T1D (**f**). **e, g** $p$-values are determined using one-sided Wilcoxon signed-rank test. The proportion of disease-specific RFUs (for CD in **e** and for T1D in **g**) among the top N significant RFUs identified by RfuWAS and TCR-seq. The red dashed lines indicate the expected proportion of antigen-specific RFUs under random prediction. Significance is assessed using Fisher's exact test, with * denoting $P<0.05$.

Glutamic decarboxylase 65, GAD65) from McPAS-TCR database[38] (Supplementary Data 6) and observed disease-associated gdRFUs were more likely to be antigen-specific, as measured by higher z-score (one-sided Wilcoxon signed-rank test, $P<0.05$; Fig. 3d, f). The proportion of antigen-specific gdRFUs in the top N significantly disease-associated pathogenic gdRFUs was higher than random (Fisher's exact test, $P<0.05$; Fig. 3e, g). In contrast, similar analysis using disease-associated RFUs from TCR-seq dataset[32,39] (Supplementary Table 3, Methods) revealed no significant difference in the proportion of antigen-specific RFUs compared to random prediction (Fig. 3e, g). Most CD-associated antigens are presented by HLA-DQ2 and T1D-associated antigens by HLA-DR4 (Supplementary Data 6). We further analyzed antigen-specific gdRFUs within HLA-restricted cohorts and observed a higher proportion of antigen-specific gdRFUs than random (Supplementary Fig. 8e, f). Together, these results potentially provided a genetic association-based approach to prioritize antigen-specific RFUs.

## gdRFU abundances associated with cancer survival

Similar to autoimmune disorders, the outcomes of most malignant cancers are strongly influenced by the adaptive immune system[40,41]. We then investigated the impact of gdRFU abundances on cancer survival using the UKBB samples. Patient survival was defined by all-cause mortality. We used Cox proportional hazard model to test the effects of gdRFU on 42 neoplasms with over 130 events, adjusted for the first ten genotype PCs, sex, and diagnosis age. We identified five significant associations (FDR < 0.05; Fig. 4a–d, Supplementary Fig. 9), comprising four protective gdRFUs and one pathogenic gdRFU. Simultaneously, we evaluated all variants within the TRB and HLA loci for their impact on cancer survival and found that none surpassed the Bonferroni-adjusted $p$-value threshold ($1.6 \times 10^{-6}$; Fig. 4a–d). Additionally, we used the gene expression of TRB and HLA genes predicted from the TWAS Elastic Net model of whole blood[42] to test associations with cancer survival. We confirmed that the associations observed for gdRFU were not covered by the survival relevance of gene expression. Our findings align with previous research suggesting that the combinations of TRBV/TRBJ gene usage and HLA alleles associated with cancer survival, whereas the impact of these elements alone was not significant[43,44]. Together, these results suggested that the combined effects from multiple variants within these loci, interpreted from an immunological perspective, could reveal new genetic factors associated with cancer survival.

We next examined the potential functions of these gdRFUs. The protective gdRFUs were likely linked to neoantigen-reactive T cells[45]. For example, gdRFU 3827, protective for lung cancer (Supplementary Fig. 9b), contained NSDHL-specific TCR[38], a gene prominently overexpressed in lung adenocarcinoma (LUAD; $P = 2.9 \times 10^{-10}$) and lung squamous cell carcinoma (LUSC; $P = 2.8 \times 10^{-22}$). It also predicted worse survival in LUAD ($HR = 1.4$, $P = 0.038$; TIMER 2.0 sever[46]). Similarly, gdRFU 3459, protective for ovarian cancer (Supplementary Fig. 9f), harbored WT1-specific TCR[47], a marker recognized for its diagnostic and prognostic significance[48,49].

We then investigated the phenotype characteristics of these protective RFUs in cancer patients. Specifically, we performed RFU analysis using the TCR sequences uncovered by TRUST[20] from over 10,000 tumor samples from the Cancer Genome Atlas (TCGA) cohort (Supplementary Table 3). By examining differentially expressed genes (DEGs) between patients with and without protective RFUs, we observed an upregulation of genes related to T cell immunity, including *CORO1A*, *ITGB2*, *CD74*, and *CXCL9*, in patients with protective RFUs (Fig. 4e). Gene set enrichment analysis[50] confirmed that genes expressed at higher levels in patients with protective RFUs were predominantly related to T cell proliferation, differentiation and activation (Fig. 4f). These results highlighted the potential role of the protective gdRFUs in fostering a T cell activation phenotype conducive to cancer protection.

## Discussion

Our study discovered significant associations between genetic variants and TCR cluster abundances, via employing a recently developed TCR embedding method. We introduced gdRFUs as a novel representation of genetically determined variation in TCR repertoires. gdRFUs allowed a systematic and unbiased investigation of whole immune repertoire association with human diseases. In addition to causal genetic variants as reported in the past GWAS studies, it is feasible now to explore the 'causal immune repertoire components' to various autoimmune disorders and cancer survival.

Building upon previous studies that investigated the associations between genetic variants and the TCR repertoire[13–16,29,51], our research uniquely employed TCR cluster abundances to assess the associations. Unlike methods that analyze V(D)J recombination phenotypes or amino acids at individual CDR3 positions[13–15], our use of RFUs synthesized these antigen recognition-related factors into a cohesive immunological interpretation. Furthermore, instead of focusing solely on shared clonotypes, which were typically limited to expanded or public clones[16,29,51], our approach aggregated clonotypes potentially with similar antigen specificity. This method was inclusive of less common clonotypes, providing better coverage of the immune repertoire. Additionally, our study extended beyond the typical focus on HLA alleles, encompassing a genome-wide perspective to explore a broader range of genetic influences on the TCR repertoire.

Our research identified a multitude of RFUs associated with a variety of diseases, including autoimmune, hematopoietic, and metabolic disorders. We substantiated their potential antigen reactivity by analyzing antigen specificity in CD and T1D and by examining T cell activity in cancer cases. Since almost no large-scale TCR-seq and phenotype datasets were currently available, it was hard to identify relevant TCR sequences for most diseases. Our study marked a pioneering effort in the phenome-wide identification of disease-associated TCR sequences. Our findings hold promise for non-invasive disease diagnostics and surveillance, building on recent successful applications of TCR repertoires in early cancer detection[17,52,53].

Importantly, our gdRFUs encapsulated the genetic effects of multiple variants, thereby not being confined to known HLA autoimmune risk alleles. The possibility of untagged causal variants or the influence of multiple causal variants on an RFU suggests that traditional colocalization analyses might miss critical signals[54]. This was exemplified in our findings of gdRFUs associated with cancer survival, where single variant associations were not apparent. The limited coverage of the TRB locus[55] further underscored the necessity of aggregating multiple rfuQTL signals. We anticipate more significant and unseen associations with RFUs when future large TCR-seq or RNA-seq samples with clinical phenotypes become available.

Despite the valuable insights into the role of TCR repertoire in genotype-phenotype associations provided, our work has several limitations. First, our predictive models only accounted for a portion of the variance in RFU abundance, likely due to the sample size of rfuQTL training set. Despite the inherent constraints imposed by the significant influence of antigen exposure on TCR repertoire diversity[16], the use of a larger cohort encompassing information from both TCR alpha and beta chains might improve both performance and heritability assessments. Second, the predominance of European ancestry in the rfuQTL dataset limited the generalizability of our findings. Given the distinct selection pressure of TCR repertoire of different populations[56], expanding our analysis to include diverse ancestries, more comprehensive genetic variants, and TCR repertoire data is crucial for future research. Third, our ability to annotate antigen-specific RFUs was limited by the currently available TCR/antigen pairs, which are both scarce and biased toward selected viral epitopes. Future research would benefit from more extensive and unbiased high-throughput identification of TCR specificity.

In conclusion, our RfuWAS analysis combined the effects of multiple variants to form an immunologically interpretable phenotype, elucidating the complex relationship between genetic variants and diseases. The RFUs associated with immune diseases and cancer outcomes not only enhanced our understanding of these conditions but might also offer opportunities in disease diagnosis and immune monitoring.

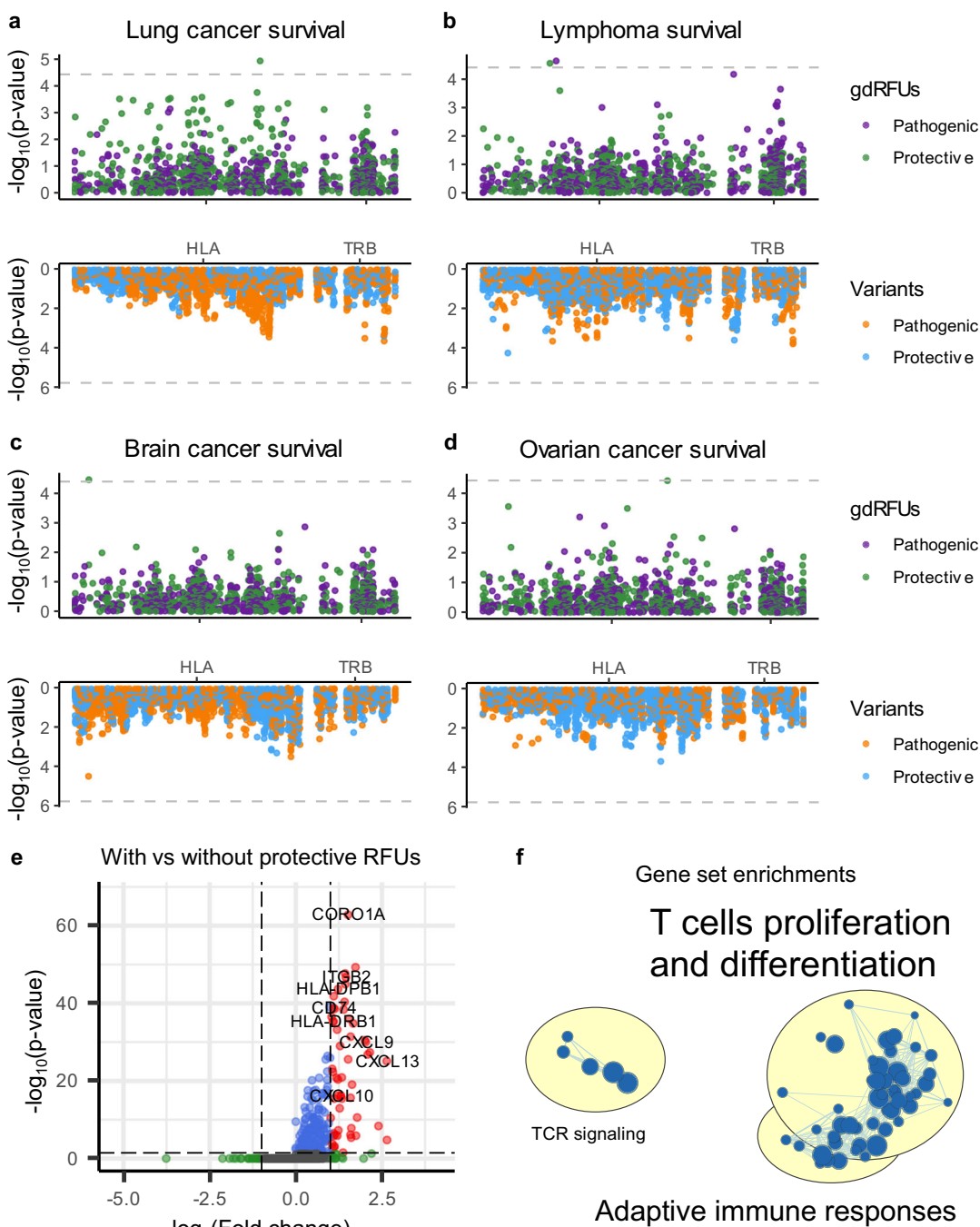

**Fig. 4 | Cancer survival outcomes and gdRFU abundances. a–d** Miami plot for cancer survival associations with gdRFU (upper) and variants in TRB and HLA loci (lower) for cancer of bronchus; lung (phecode 165.1, **a**), diffuse large B cell lymphoma (phecode 202.24, ICD 10 code C83.3, **b**), malignant and unknown neoplasms of brain and nervous system (phecode 191, **c**), malignant neoplasm of ovary and other uterine adnexa (phecode 184.1, **d**). Each point represents a *p*-value of the Cox proportional hazards models with gdRFU or genetic variants in HLA and TRB loci as variables, ordered by genomic position and colored by pathogenic or protective effects. The genomic position of gdRFU is defined as the variant position with the most significant association. Gray dashed lines indicate a Bonferroni-corrected p-value significance threshold. **e** a volcano plot showing genes differentially expressed between patients with protective RFUs and without protective RFUs. Fold changes were calculated using the median values of the two groups, and *p*-value were determined using the Wilcoxon signed-rank test and adjusted with Bonferroni procedure. The horizontal dashed line represents FDR of 0.05, and the vertical dashed line indicates a fold change of 2. Genes are color-coded as follows: red for significant differential expression, green for fold change > 2 or < ½ with FDR > 0.05, blue for fold change between ½ and 2 with FDR < 0.05, and grey for fold change between ½ and 2 with FDR > 0.05. **f** a gene set network diagram for gene sets enriched in patients with protective RFUs. Each node represents a gene set, with the node size correlating to the number of genes. For visualization clarity, only gene sets with an $FDR < 5 \times 10^{-5}$ were shown, and four gene sets (related to chemokine response and leukocyte apoptotic processes) distant from the main cluster were omitted.

## Methods

### The rfuQTL training set

The WGS and RNA-seq data for the rfuQTL training set were downloaded from the database of Genotypes and Phenotypes (dbGaP) under accession number phs001442. We identified TCR sequences from RNA-seq data employing TRUST4[20], aggregated TCR sequences from multiple time points per individual, and calculated RFU abundance for each individual. Since the dataset was an RNA-seq dataset with lower coverages of TCR sequences

compared with TCR-seq, we did not filter the dataset. Ancestry inference was conducted using GRAF-pop[57].

The variant array data were filtered to retain only those variants exhibiting a minor allele frequency (MAF) above 0.05 and Hardy-Weinberg equilibrium $p$-value (pHWE) above $10^{-6}$. This filtering yielded a total of 6,390,032 variants. We excluded individuals who deviated by more than $\pm 3$ standard deviation in heterozygosity rate from the mean, those exhibiting high relatedness (defined as a Kinship-based INference for Genome-wide association studies (KING) coefficient greater than 0.0884), and individuals with fewer than 2500 unique CDR3 beta chains. The number of 2500 was chosen to remove outliers in the first two PC of RFU abundances (Supplementary Fig. 2c, d). HLA haplotypes were imputed from the WGS variant data using Michigan Imputation Server[58] with the Four-digit Multi-ethnic HLA v1 reference panel[59]. Only HLA haplotypes with a frequency greater than 0.05 were included in subsequent analyses.

### The rfuQTL test set
Genotype data for the rfuQTL test set were downloaded from dbGaP (phs001918), the TCR-seq data were downloaded from the ImmuneAccess database (DOI:10.21417/B7001Z), and the HLA haplotype data were retrieved from https://github.com/immunogenomics/cdr3-QTL. The dataset's genotype imputation and quality control have been detailed in a prior publication[60]. For each individual, we calculated the RFU abundance from the 20,000 TCR sequences exhibiting the highest clonality ("seq_reads" column in the processed files) in the TCR-seq data. We selected 20,000 TCR sequences to make the number of sequences comparable to the rfuQTL training set, as 20,000 was proximate to the mean plus one standard deviation of the training set. Inclusion criteria for HLA haplotypes were identical to those of the training set, with a frequency threshold of greater than 0.05.

### The RfuWAS dataset
A case-control study for each phecodes[61] was constructed using International Classification of Diseases (ICD-9 and ICD-10) codes from UKBB, facilitated by the tool available at https://github.com/umich-cphds/createUKBphenome. We selected phecodes with a prevalence exceeding 0.1%. The study cohort was restricted to unrelated individuals of white British ancestry, with unrelatedness defined as used in principal component analysis (PCA) calculation. Additional inclusion criteria included the absence of putative sex chromosome aneuploidy and the availability of genotype data. We retained only those variants within the TRB and HLA loci that overlapped with the rfuQTL training set and exhibited an imputation quality score greater than 0.8, a call rate exceeding 0.95, a MAF above 0.001, and a pHWE above $10^{-10}$. Imputed HLA haplotypes were binarized with 0.7 as threshold.

### RFU annotations
**CD4 and CD8 annotations**. RFU abundances for CD4 and CD8 T cells were calculated for each individual using data from two sorted TCR-seq datasets[27,28]. Cell type annotation was based on consistent enrichment observed in both datasets, as determined by the paired Wilcoxon signed-rank test ($P<0.05$).

TN, CM, Treg, and Tscm annotations: RFU abundances for each T cell subset in individuals were calculated using a sorted TCR-seq dataset[32]. For predictable RFUs, differences among T cell subsets were assessed using the Friedman test, applying a Bonferroni-corrected $p$-value threshold ($P<0.05/1351 = 3.7 \times 10^{-5}$). The cell type with the highest median abundance was designated as the enriched cell type.

**Antigen-specific RFUs**. Antigen-specific TCR sequences were obtained from the McPAS-TCR database[38], identified either through peptide-MHC tetramers or in vitro stimulation, and the VDJdb database[47]. These sequences were then mapped to our RFUs. An RFU was classified as "antigen-specific" if it included any of these sequences. We assigned the

RFU as specific to all antigens if more than one antigen-specific TCR mapped to it. Since many TCR sequences reacted to multiple antigens due to their known cross-reactivity[62], it was normal to have more than one antigen specificity for a given RFU. Since the numbers of TCR sequences specific to CD and T1D were limited in VDJdb compared to McPAS-TCR, we only used McPAS-TCR for CD and T1D-specific TCRs. Both databases were used for annotating gdRFUs associated with cancer survival.

### rfuQTL for whole-genome variants and HLA haplotypes
RFUs with an abundance above zero in at least 20% of individuals were selected, resulting in 4953 for the rfuQTL training set, and 4974 for the test set. The distribution of RFU abundance in each individual was quantile normalized to the average empirical distribution across all individuals. The abundance for each RFU was subsequently quantile normalized to the standard normal distribution.

For the training set, sex, the first five genotype PCs, which were associated with the source of individuals (Supplementary Fig. 3a–c), and the first two probabilistic estimation of expression residuals (PEER)[23] factors, explaining a large portion of RFU abundance variance (Supplementary Fig. 3d, e), were used as covariates. For the test set, the first three HLA PCs (as in *Ishigaki et al.*[15]) and the first eight PEER factors were utilized as covariates, since these PEER factors associated with potential confounders and explained substantial variance in RFU abundance (Supplementary Fig. 3f, g). Direct use of these potential covariates were avoided due to missing values.

Covariates were first regressed out from RFU abundance to obtain PEER residuals using PEER. The association between these residuals and variants or HLA haplotype was then tested using a linear model with Matrix eQTL[63]. To control for multiple testing, a Bonferroni correction was applied, setting the significant threshold at $P<5 \times 10^{-8}/4953 = 1.0 \times 10^{-11}$ for whole-genome association analysis and $P<0.05/28/4953 = 3.6 \times 10^{-7}$ for HLA haplotype analysis in rfuQTL training set.

### Function analysis of rfuQTL variants
The Variant Effect Predictor (VEP)[64] was utilized to annotate the consequences of the identified variants. The impact of missense variant rs17229 on TRBV12-5 was annotated using IMGT/V-QUEST[65].

To examine regulatory elements, we downloaded ENCODE cCREs from the UCSC Genome Browser, under the group "Regulation", track "ENCODE cCREs", and file "encodeCcreCombined.bb". We downloaded ENCODE transcription factor binding site data from the same browser and group, under the track "TF clusters", and file "encRegTfbsClusteredWithCells.hg38.bed.gz". We assessed the enrichment of rfuQTL variants in both tracks compared to all variants overlapping each track, applying a one-sided Fisher Exact Test. For transcription factor enrichment analysis, we used the Bonferroni-corrected significant threshold at $P < 0.05/324 = 1.5 \times 10^{-4}$.

### Lasso model training and testing
In rfuQTL training set, we developed both lasso and elastic net models for each variant set and each RFU to predict normalized and covariates-corrected RFU abundances. The input of the model is the genotypes of all subjects, and the output is the RFU abundance for them. Since lasso models and Elastic Net consistently achieve strong performance in similar tasks, we began with these two models and chose lasso model since it performed better. Variants in rfuQTL training set were converted from GRCh38 to GRCh37 using the Ensembl Assembly Converter[66]. Variant sets examined included those in the TRB locus, HLA locus, and combined TRB + HLA locus. For each locus, variants within 1 Mb window, variants specific to the gene locus (denoted with "o"), and variant windows defined by the first and last variant showing significant associations with at least one RFU ($P<5 \times 10^{-8}$; denoted with "s") were used. Given the similar and robust performances of the TRB and TRB + HLA variant sets, the best-performing model for each RFU was chosen from these sets.

Narrow-sense heritability was calculated as an upper limit of our models. The genetic relationship matrix (GRM) was generated from variants in each variant set, and a restricted maximum likelihood (REML) analysis with GCTA was performed to estimate the variance explained by these variants. All variance estimations were constrained to be positive, ensuring $h^2$ comparability with $R^2$.

The lasso model was utilized to predict RFU abundances in the rfuQTL test set (n = 398). For variants present in the lasso model but absent in the test set, their variant count was assigned as NA. PCs 1-8 (as in Russell et al.[13]) and PEER factors 1-8 were used for normalized RFU abundance correction. Model performance was evaluated by comparing corrected RFU with predicted RFU using available prediction, and heritability estimates from the test set served as performance upper limits.

We found models using variants overlapping between both the training set and UKBB performed similarly to those using all variants in the training set, while models using the overlapping variants among the training set, test set, and RfuWAS performed worse (Supplementary Table 4). Thus, we adopted the model using variants overlapping between the training set and RfuWAS in this work.

### RfuWAS analysis
The lasso model was applied to predict gdRFUs using genetic variants in the RfuWAS dataset. Covariates incorporated in the analysis included first ten genotype PCs, inferred sex, and age at recruitment. We used PrediXcan[31] to first regress out all covariates from the phenotype data, and then examined the associations between gdRFU abundances and covariate-corrected phenotypes. The Bonferroni corrected $p$-value threshold was set at $P < 0.05/1086/1351 = 3.4 \times 10^{-8}$. Diseases listed on https://autoimmune. org/disease-information/ were classified as autoimmune diseases in annotation.

### Prioritization of antigen-specific RFUs with TCR-seq dataset
RFUs were calculated from CD TCR-seq and T1D TCR-seq datasets using the first 10,000 abundant TCR sequences (the "max_productive_frequency" column in ImmuneAccess files), which is the common practice for TCR-seq data analysis. Associations between diseases and RFUs were assessed using linear regression. Antigen-specific gdRFUs were prioritized based on their $p$-values from pathogenic RFUs in the TCR-seq datasets.

### Cancer survival analysis
Cancer record from UKBB category 100092 and death record from category 100093 were used. All-cause mortality served as the endpoint in constructing Cox proportional hazard models to evaluate associations between gdRFU abundances or genetic variants and neoplasm survival ($n = 42$). The first ten PCs, inferred sex, and diagnosis age were included as covariates. Associations were assessed only for cancers with a minimum of 130 cases. The threshold for significant association was set at $FDR < 0.05$.

### Phenotype characteristics of protective RFUs analysis
RFU abundances were calculated from TCR sequences identified in RNA-seq data from TCGA. We focused on the 2000 genes with the highest variances to identify the DEGs between patients with and without protective RFUs. GSEA was performed on the expression of all genes in Gene Ontology gene sets, followed by visualization with EnrichmentMap. AutoAnnotate was employed for automatic gene set cluster annotation, with subsequent manual refinement of cluster names to improve annotation accuracy[67].

### Statistics and Reproducibility
Details of statistical analyses and sample sizes were described in the Methods section and figure legends. We used R 4.2.3 (https://www.r-project.org/) for statistical analyses.

### Reporting summary
Further information on research design is available in the Nature Portfolio Reporting Summary linked to this article.

## Data availability
Raw data analyzed in this study are available at the following locations: dbGaP: phs001442, phs001918; ImmuneAccess database: https://doi.org/ 10.21417/B7001Z, https://doi.org/10.21417/B7H01M, https://doi.org/10. 21417/B7C88S, https://doi.org/10.21417/LWL2022JCP. Access to UK Biobank individual-level data can be requested from https://www. ukbiobank.ac.uk/enable-your-research/apply-for-access. The weights of the lasso models are deposited at GitHub: https://github.com/YuhaoTan2/ RfuWAS/blob/main/models/lasso_weights_tsv.zip.

## Code availability
All code used in the study is available at https://github.com/YuhaoTan2/ RfuWAS and deposit at Zenodo (https://doi.org/10.5281/zenodo. 13646450)[68]. Other software used includes TRUST4; GRAF-pop 1.0 (https://www.ncbi.nlm.nih.gov/projects/gap/cgi-bin/GetZip.cgi?zip_ name=GrafPop1.0.tar.gz); plink 2.0 (https://www.cog-genomics.org/plink/ 2.0/); Michigan Imputation Server 1.2.4 (https://imputationserver.sph. umich.edu/index.html#!); PEER 1.3 (https://github.com/PMBio/peer.git); GCTA 1.94.1 (https://yanglab.westlake.edu.cn/software/gcta/bin/gcta-1.94. 1-linux-kernel-3-x86_64.zip); CreateUKBphenome (https://github.com/ umich-cphds/createUKBphenome); MatrixEQTL 2.3 (https://cran.r- project.org/web/packages/MatrixEQTL/index.html); PrediXcan 0.6.11 (https://github.com/hakyimlab/MetaXcan); GSEA 4.3.2 (https://www.gsea- msigdb.org/gsea/index.jsp); EnrichmentMap 3.3.5 (https:// enrichmentmap.readthedocs.io/en/latest/); AutoAnnotate 1.4 (https:// autoannotate.readthedocs.io/en/latest/).

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

## Acknowledgements
This work is supported by NCI R01 CA258524 (B.L.), CA245318 (B.L.), NIAID U01 AI169298 (X.Z.), R01 AI174108 (D.J.L.), U01 AI185638 (D.J.L.), and NHGRI R01 HG011035 (D.J.L.). We acknowledge Philip Bradley for discussion on the project.

## Author contributions
B.L., X. Z., D.J.L., and Y.T. conceived the project. Y.T. developed the framework and performed analysis. L.W., H.Z., M.P., B.L., X. Z., and D.J.L. helped with data interpretation. B.L., X.Z., D.J.L., and Y.T. facilitated dataset access. Y.T., B.L., X.Z., and D.J.L. prepared the manuscript. B.L., X.Z., and D.J.L. supervised the study.

## Competing interests
The authors declare no competing interests.
