## [Peer Review File · Communications Biology]

Reviewers' comments:

Reviewer #1 (Remarks to the Author):

In this manuscript, Tan et al. performed genetic association between T cell receptor (TCR) and found significant associations at TRB and HLA loci. They then developed statistical model to predict TCR components and applied such models to UK Biobank to perform systematic disease associations. Their results suggested significant enrichment for autoimmune diseases and identified TCR clusters associated with cancer survival. The overall study was well designed with interesting results, but it's unclear to me what additional insights this study could provide. I have the following specific comments.

1. (major) The biggest concern I have is the additional value of studying RFU abundance over just the HLA and TRB loci, especially given the models of RFU prediction only included variants in these two loci, and the two loci are also well-established to be associated with autoimmune diseases.

- What insights do the authors have?

- Are there any additional disease associations in your rfuWAS results compared to usual GWAS/PheWAS associations?

- In the cancer survival analysis, the authors compared results from predicted gdRFU abundance versus usual GWAS results. How about results from gene expressions/usual TWAS models for TRB and HLA genes?

2. (major) In the GWAS of RFU abundance, the authors mentioned they found 2,083 associations, but it seems those associated variants are not independent. Some of them are in high LD, which would be misleading to report the numbers. How many independent/distinct signals discovered? Any other suggestive loci beyond TRB and HLA?

3. (major) The authors studied cell type specificity of RFU, but it's not clear to me whether the authors accounted for cell type proportion in RFU association and prediction.

4. (major) The author assessed disease-associated gdRFUs in different cell types. How about the gene expressions of HLA and TRB genes among different cell types? Do they exhibit similar patterns?

5. (minor) The authors mentioned that they calculated heritability from GCTA using variants in HLA and TRB locus. Did the authors calculate SNP heritability using genome-wide variants? How are the two h^2 's comparable?

Reviewer #2 (Remarks to the Author):

In this work, Tan and colleagues extend upon the GIANA T cell receptor sequence clustering model to identify putative links between genotypic variants and T cell receptor clusters, defined as Repertoire Functional Units (RFUs). Having established correlations between genotype (specifically between the HLA and TRB loci) and RFU makeup, the authors assess whether genotype can be leveraged to predict RFU abundance, and so to investigate the correlation between predicted RFUs and other features including T cell functional subtype, autoimmunity, and cancer outcomes.

Major comments:

1. A core concept in this study is the use of RFUs as predictive markers of TCR antigen specificity. The definition of an RFU refers to a study in which GIANA is used to cluster TCRs drawn from a large public dataset of unpaired TCRs, and the resultant clusters are used to generate trimer substitution matrices as a feature for clustering on the basis of common antigen specificity. Whilst GIANA has been shown to outperform comparators in certain circumstances, the reliability of RFUs as a method for generalisable inference of TCR specificity has not been confirmed beyond reasonable doubt. Specifically, the publication in which RFUs are proposed:

A) Tests the new approach on a single TCR dataset of 10 epitopes under a single pre-processing approach, when independent benchmarking has shown that performance of TCR clustering algorithms is highly sensitive to the dataset, the antigen, chain selection, and the pre-processing approach <https://doi.org/10.1016/j.immuno.2024.100033>

B) Achieves 64% ROC AUC on this dataset (where recent predictive models achieve >70% for seen epitopes)

C) Does not contextualise performance with any other clustering approach or with an appropriate (for example random) baseline

D) Is available as a preprint, and thus does not appear to have been subject to formal peer review

I would therefore strongly encourage that the authors include evidence to support the generalisability of the RFU clustering approach to other datasets and epitopes, including with sensitivity to pre-processing parameters, and compare performance with relevant

comparator and baseline models. This is particularly important given the conclusions drawn for other associations hinge on the validity of the RFU clustering approach. An experimental test, for example that orphan TCRs falling into the same RFU as a TCR of known specificity have the same specificity, would add further confidence.

Minor comments:

1. Line 20: It would be helpful to explain what is meant by TCR components in the abstract, for example the frequency of clusters of related TCR sequences predicted to have common antigen specificity
2. Line 25: Enriched for predicted antigen specificity, as antigen specificity has not been confirmed but is posited based on co-clustering with labelled TCRs
3. Line 48: Can the authors explain how RFUs differ from a bag of kmers? It would be helpful to have a schematic of the RFU extraction pipeline (e.g. similar to that in reference 18), as this is an important concept for the rest of the article
4. Line 67: Does aggregating across time points skew the dataset towards patients who are sampled at multiple time points?
5. Line 72-4: to what extent does normalisation and PEER adjustment improve performance of the regressor?
6. Line 75: Is the correlation between TRB and RFU not exactly what one would expect, given encoding of TCR germline sequences that will be reflected in the N terminus of the CDR3 (line 80)? What is the novelty of this, and what happens to the significance of non-TCR coding genes when you mask the TRB locus?
7. Fig. 1d a smaller dot size on the scatterplot and jitter might make this plot easier to interpret (as for other figures)
8. Line 185: The assertion that TCRs assigned to the same RFU will have a 90% chance of having the same antigen specificity is not based on strong evidence, as set out above, particularly given the dataset in question is pre-processed to remove TCRs having the same sequence but different antigen specificity
9. Line 187: Were there any specific findings for Antigen-specific TCRs drawn from VDJdb (mentioned in the methods)?
10. Line 328-9: What was the proportion of exact matches between the training and test sets, and how was clonality determined?
11. Line 353-4: How did you handle cases (if any) in which more than one antigen-specific TCR mapped to a given RFU?
12. Line 402-405: The methodology for feature selection and train/test validation is not clear. Please provide a concise description of the feature vector that is provided to the model, how and why a lasso model was selected, performance of an appropriate

comparator model, and results when there is no overlap between the train and test sets.
Line 414: Why was a different threshold used for the TCR-seq dataset than McPas-TCR/VDJdb?

Reviewers' comments:

Reviewer #1 (Remarks to the Author):

In this manuscript, Tan et al. performed genetic association between T cell receptor (TCR) and found significant associations at TRB and HLA loci. They then developed statistical model to predict TCR components and applied such models to UK Biobank to perform systematic disease associations. Their results suggested significant enrichment for autoimmune diseases and identified TCR clusters associated with cancer survival. The overall study was well designed with interesting results, but it's unclear to me what additional insights this study could provide. I have the following specific comments.

We thank the reviewer for the positive and constructive comments. We have extensively revised the manuscript accordingly.

1. (major) The biggest concern I have is the additional value of studying RFU abundance over just the HLA and TRB loci, especially given the models of RFU prediction only included variants in these two loci, and the two loci are also well-established to be associated with autoimmune diseases.

- What insights do the authors have?

- Are there any additional disease associations in your rfuWAS results compared to usual GWAS/PheWAS associations?

- In the cancer survival analysis, the authors compared results from predicted gdRFU abundance versus usual GWAS results. How about results from gene expressions/usual TWAS models for TRB and HLA genes?

Thank you for bringing up the concern.

(1) Insights: Although variants in HLA and TRB loci are known to be associated with autoimmune diseases, the mechanism underlying the associations was unclear. Previous research has indicated that these associated HLA risk alleles shape TCR repertoire during thymic selection¹. However, these analyses were limited to three autoimmune diseases (celiac disease, type 1 diabetes, and rheumatoid arthritis), without covering the effects in other diseases. Besides, they only examined the effects of known autoimmune risk alleles, but TCR repertoires were influenced by both risk variants and other variants. As a result, the role of TCR repertoire in explaining genotype-phenotype associations has not been fully explored.

In our study, we provided more evidence for the hypothesis that genetic variants influence disease risks through impacting TCR repertoires. Instead of only focusing on the known autoimmune risk variants, we combined all variants related to TCR cluster abundance to explain the genotype-phenotype associations. We found that the associations between mutations and various diseases, including autoimmune, hematopoietic, and metabolic disorders, could be explained by the associations between mutations and the TCR

repertoire. Besides, our model identified variant sets associated with cancer prognosis that could not be detected by conventional GWAS, revealing the ability of our method to uncover novel genetic associations.

In summary, by studying RFU abundance beyond just the two loci, we identified the role of TCR repertoire in explaining the effects of variants on multiple diseases and cancer survival. We edited our introduction to highlight these insights:

“A recent study provided genetic evidence supporting the hypothesis that HLA risk alleles shape T cell repertoire during thymic selection¹. However, their association analysis was limited to three autoimmune diseases (celiac disease (CD), type 1 diabetes (T1D), and rheumatoid arthritis (RA)), without covering the effects in other diseases. Besides, they examined only known autoimmune risk variants, whereas TCR repertoires could be influenced by other variants. Additionally, the study used amino acid compositions of the CDR3 repertoire for quantification, which are not directly related to antigen specificity and T cell phenotypes. To address these limitations, we used the recently developed embedding method, Repertoire Functional Units (RFUs)², for TCR repertoire quantification. This method divides the TCR space into a fixed number of RFUs, each representing a cluster of TCRs with similar sequences. To quantify a given TCR repertoire, we assigned each TCR sequence to an RFU by identifying the nearest neighbor in the embedding space, and used the abundance of each cluster as a numerical encoding of the repertoire (Supplementary Fig. 1).

In this study, we provide more direct evidence to support the role of TCR repertoires in variant-disease associations leveraging UK Biobank (UKBB)³, a large cohort with extensive clinical information. We developed a lasso-based model to predict RFU abundances based on genetic variants. Applying this model to UKBB samples, we predicted RFU abundances and discovered their associations with various diseases, including autoimmune, hematopoietic, and metabolic disorders. Besides, we identified variant sets associated with cancer prognosis that could not be detected by conventional GWAS. Our novel analytical framework provides a systematic approach to understand the impact of TCR repertoires across the phenome, presenting them as immunologically meaningful variant aggregates to clarify the complex links between genotypes and a broad spectrum of diseases.”

- (2) Additional disease associations: We compared our UKBB RfuWAS results with PheWAS associations from the Pan UK Biobank⁴. We identified that Calculus of kidney (Phecode 594.1) was associated with RFU 2579 ($p = 7.1 \times 10^{-10}$) but no variants in the TRB and HLA loci were associated with this condition. T cells have been implicated in the formation of kidney stones⁵, and it is possible that selected T cell clones are related to the disease. We included this analysis in our results:

“By comparing our results with GWAS associations in the Pan UK Biobank⁴, we identified that the kidney calculus (Phecode 594.1) was associated with RFU 2579 ($p = 7.1 \times 10^{-10}$) but no variants in the TRB and HLA loci were associated with this condition. T cells have been implicated in the formation of kidney stones⁵, and it is possible that selected T cell clones are related to the disease.”

(3) Comparison with TWAS in cancer survival: We predicted the gene expression of TRB and HLA genes for the UKBB cohort with the TWAS Elastic Net model of whole blood. Only 16 HLA genes and TRBV11-2 had corresponding models. We assessed the associations between predicted gene expression and cancer survival, and observed one significant association between HLA-A and prostate cancer (Phecode 185, $p=0.0015$). Our RfuWAS results did not overlap with TWAS results, indicating that the associations we observed were independent of gene expression. We added the TWAS analysis to our revised manuscript:

“Additionally, we used the gene expression of TRB and HLA genes predicted from the TWAS Elastic Net model of whole blood to test associations with cancer survival. We observed one association between HLA-A and prostate cancer ($p=0.0015$), which did not overlap with our gdRFU associations, indicating that our associations were independent of gene expression.”

2. (major) In the GWAS of RFU abundance, the authors mentioned they found 2,083 associations, but it seems those associated variants are not independent. Some of them are in high LD, which would be mis-leading to report the numbers. How many independent/distinct signals discovered? Any other suggestive loci beyond TRB and HLA?

Thank you for pointing out the LD between variants. To identify independently associated variants for each RFU, we iteratively included the most significantly associated variants as covariates for linear regression until no more significant signals existed. We found 105 independent associations for 59 RFUs.

With the stringent genome-wide p-value threshold across all RFUs ($p < 5 \times 10^{-8}/4953$), we did not observe any significant loci beyond TRB and HLA. However, using the p-value threshold for each RFU (5×10^{-8}), we observed 2,245 significant associations beyond these two loci. Among them, we found 93 variants in the ZNF443 locus and 6 variants in the ZNF709 locus associated with RFU 3490. These two genes have been reported to be associated with TRBV24-1 gene usage frequency of TCR sequences⁶. Some TCR sequences belonging to RFU 3490 contained the initial sequence of TRBV24-1 (“CATSDL”), revealing that the effects of these two genes may be due to their effects on gene usage. Besides, we observed 11 variants near the LIG4 locus associated with RFU 3970, a gene essential for V(D)J recombination through nonhomologous end joining. We also observed associations related to signal transduction (GNAS, STK38, GNG2) and transcription regulation (RBM20), which might be associated with the TCR repertoire.

3. (major) The authors studied cell type specificity of RFU, but it's not clear to me whether the authors accounted for cell type proportion in RFU association and prediction.

We followed the reviewer's suggestion and analyzed the potential effects of cell type proportion in RFU associations. We first assigned each RFU with the proportion of CD4, CD8, CM, TN, Treg, Tscm cell types based on the TCR-seq datasets with sorted T cells. We then calculated the cell type proportions for each sample by summing RFU abundance weighted by the proportion of each cell type. We did not observe significant differences in cell type proportions across subjects. After correcting for CD4, CM, TN, and Tscm cell type proportion

in the regression model, we found that the RFU abundances were almost the same as in the original analysis: The mean squared error of RFU abundance between the current analysis and original analysis was only 5×10^{-6} , and the mean Pearson correlation for each sample was 1.0. We then assessed the associations between variants and corrected RFU abundances, and found 2,084 significant associations, with 2,083 associations overlapping with the original analysis. The t statistics of associations that both $p < 5 \times 10^{-8}$ was almost perfectly correlated (Fig. R1), revealing that cell type proportions have limited impacts on variant-RFU associations in rfuQTL training set. We added the analysis in the revised manuscript:

“Adjusting for cell type proportions did not affect the association results (Supplementary Fig. 4b).”

Fig. R1 | The t-statistics of associations between genetic variants and RFU abundances with cell type proportions corrected and not corrected.

4. (major) The author assessed disease-associated gdRFUs in different cell types. How about the gene expressions of HLA and TRB genes among different cell types? Do they exhibit similar patterns?

Thank you for raising this point. We tested the associations between the expression of TRB and HLA genes and the cell type proportions in rfuQTL training set. We used the 771 samples with at least 2,500 TCR sequences and used the same methods as in our response to comment 3 to calculate cell type proportions. We observed 23 significant associations after Bonferroni correction (Table R1).

We then tested the proportion of RFUs of each cell type that were associated with diseases. Since CD8 T cells recognize antigens presented by MHC-I, and CD4 T cells recognize antigens presented by MHC-II, we analyzed the fraction of MHC-I and MHC-II genes that were associated with diseases. We assessed the associations between diseases and predicted gene expressions as mentioned in response to comment 1. We calculated the fractions of MHC-I and MHC-II genes associated with each disease (Fig. R2), and compared them with the fractions of CD4 and CD8 gdRFUs in Supplementary Fig. 8b. Some diseases had similar

patterns, for example, a large fraction of both CD4 gDRFUs and MHC-II expressions were associated with Type 1 diabetes, whereas some diseases exhibited different patterns.

Table R1. Associations between cell type proportion and gene expression.

Cell type proportion	Gene	Pearson correlation	p-value	Adjusted p-value
TN	TRBJ1-6	0.468	2.55e-43	1.55e-40
Treg	TRBJ1-6	-0.368	4.10e-26	2.50e-23
Treg	TRBJ2-7	-0.359	6.20e-25	3.78e-22
TN	TRBJ2-7	0.342	1.38e-22	8.43e-20
TN	TRBV30	0.262	1.56e-13	9.51e-11
Treg	TRBV30	-0.230	9.82e-11	5.99e-08
CM	TRBJ1-6	-0.199	2.38e-08	1.45e-05
CD4	TRBJ2-7	-0.180	5.12e-07	3.12e-04
Treg	HLA-DRB5	-0.179	5.90e-07	3.60e-04
TN	TRBJ2-1	0.175	1.08e-06	6.61e-04
Treg	TRBJ2-1	-0.173	1.38e-06	8.40e-04
Tscm	TRBJ2-7	0.166	3.44e-06	2.10e-03
TN	TRBV29-1	0.166	3.72e-06	2.27e-03
CD4	TRBV4-2	-0.160	8.57e-06	5.23e-03
TN	TRBC2	0.156	1.36e-05	8.27e-03
CD4	TRBV2	0.154	1.85e-05	1.13e-02
TN	HLA-DRB5	0.152	2.31e-05	1.41e-02
Treg	TRBC2	-0.147	4.14e-05	2.53e-02
CM	TRBJ2-7	-0.146	4.60e-05	2.80e-02
CM	TRBV28	-0.144	5.76e-05	3.52e-02
TN	TRBV27	0.142	7.18e-05	4.38e-02

Fig. R2 | The fraction of disease associated genes within MHC-I and MHC-II genes. The numbers of associated genes are indicated in parentheses following each disease type, and the numbers of genes per gene type are indicated in parentheses following each gene type.

5. (minor) The authors mentioned that they calculated heritability from GCTA using variants in HLA and TRB locus. Did the authors calculate SNP heritability using genome-wide variants? How are the two h^2 's comparable?

We followed the reviewer's suggestions to calculate the SNP heritability using genome-wide variants with GCTA on the rfuQTL training set. We found the whole genome h^2 was much higher than that of the TRB and HLA loci (Fig. R3). However, heritability estimations of whole-genome variants with only 659 samples were not accurate: the median standard error was 0.37 for whole-genome variants, whereas it was 0.018 for variants at the TRB and HLA loci. Thus, the heritability estimation might not be accurate enough, so we did not report the whole-genome heritability in the manuscript.

Fig. R3 | The heritability estimation for whole-genome variants and variants at the TRB and HLA loci in rfuQTL training set.

Reviewer #2 (Remarks to the Author):

In this work, Tan and colleagues extend upon the GIANA T cell receptor sequence clustering model to identify putative links between genotypic variants and T cell receptor clusters, defined as Repertoire Functional Units (RFUs). Having established correlations between genotype (specifically between the HLA and TRB loci) and RFU makeup, the authors assess whether genotype can be leveraged to predict RFU abundance, and so to investigate the correlation between predicted RFUs and other features including T cell functional subtype, autoimmunity, and cancer outcomes.

We thank the reviewer for carefully reading through our manuscript and for accurately interpreting our results. We have made extensive revisions to the manuscript according to the reviewer's comments and suggestions.

Major comments:

1. A core concept in this study is the use of RFUs as predictive markers of TCR antigen specificity. The definition of an RFU refers to a study in which GIANA is used to cluster TCRs drawn from a large public dataset of unpaired TCRs, and the resultant clusters are used to generate trimer substitution matrices as a feature for clustering on the basis of common antigen specificity. Whilst GIANA has been shown to outperform comparators in certain circumstances, the reliability of RFUs as a method for generalisable inference of TCR specificity has not been confirmed beyond reasonable doubt. Specifically, the publication in which RFUs are proposed:

A) Tests the new approach on a single TCR dataset of 10 epitopes under a single pre-processing approach, when independent benchmarking has shown that performance of TCR clustering algorithms is highly sensitive to the dataset, the antigen, chain selection, and the pre-processing approach <https://doi.org/10.1016/j.immuno.2024.100033>

B) Achieves 64% ROC AUC on this dataset (where recent predictive models achieve >70% for seen epitopes)

C) Does not contextualise performance with any other clustering approach or with an appropriate (for example random) baseline

D) Is available as a preprint, and thus does not appear to have been subject to formal peer review

I would therefore strongly encourage that the authors include evidence to support the generalisability of the RFU clustering approach to other datasets and epitopes, including with sensitivity to pre-processing parameters, and compare performance with relevant comparator and baseline models. This is particularly important given the conclusions drawn for other associations hinge on the validity of the RFU clustering approach. An experimental test, for example that orphan TCRs falling into the same RFU as a TCR of known specificity have the

same specificity, would add further confidence.

We appreciate your thorough review and suggestions regarding our RFU method. Though it was initially a preprint, it has recently been accepted by Cell Reports Medicine⁷. Our RFU method divided the TCR space into 5,000 RFUs, and each RFU represented a cluster of TCRs with similar antigen specificity. To quantify a given TCR repertoire, we calculated the embedding of each TCR sequence, assigned them to an RFU by finding the nearest neighbor in the embedding space, and used the abundance of each cluster as a numerical encoding of the TCR repertoire.

To demonstrate generalizability, we repeated our benchmark on another 896 TCR sequences specific to ten randomly selected epitopes (Supplementary Table 10). For each pair of TCR sequences, we calculated the Euclidean distance of their embedding as the predictor, with the response being if or not the two TCRs share the same antigen. We achieved an AUROC of 0.65, which is similar to the 0.64 in the manuscript (Fig. R4a). It revealed that our method generalized well to other datasets and epitopes. For the preprocessing parameters, we selected epitopes randomly while avoiding using some of the heavily profiled antigens. We maintained the filtering to avoid bias.

We compared our method to GIANA and hamming distance baseline. GIANA was the state-of-the-art method in the benchmark paper, and hamming distance was a baseline method. We used the Euclidean distance between the embedding of GIANA or the hamming distance between sequences to predict whether a pair of TCR sequences was specific to the same antigen. These two methods could only calculate distances for TCR sequences with the same length, and we used sequences with lengths 13, 14, and 15 since there were at least 150 sequences with the selected length. The AUROC ranged from 0.56 to 0.71 for GIANA (Fig.R4b) and ranged from 0.45 to 0.7 for hamming distance (Fig.R4c), revealing that our RFU achieved similar performance as GIANA and performed better than hamming baseline.

We included a description of the RFU method in our introduction and the new benchmark in our results.

“This method divides the TCR space into a fixed number of RFUs, each representing a cluster of TCRs with similar sequences. To quantify a given TCR repertoire, we assigned each TCR sequence to an RFU by identifying the nearest neighbor in the embedding space, and used the abundance of each cluster as a numerical encoding of the repertoire (Supplementary Fig. 1). ...

To demonstrate that our previous benchmark results were generalizable to diverse antigens, we repeated the benchmark with ten randomly selected epitopes. We found that the RFU method achieved similar performance, revealing its robustness across different antigens (Supplementary Table 10, Supplementary Fig. 8b).”

Fig. R4 | The comparison of RFU, GIANA, and hamming distance baseline on benchmark dataset.

Updated Supplementary Fig. 1 | Study design.

Minor comments:

1. Line 20: It would be helpful to explain what is meant by TCR components in the abstract, for example the frequency of clusters of related TCR sequences predicted to have common antigen specificity

Thank you for the suggestion. We added the description of TCR components in the abstract “the frequency of clusters of TCR sequences predicted to have common antigen specificity”.

2. Line 25: Enriched for predicted antigen specificity, as antigen specificity has not been confirmed but is posited based on co-clustering with labelled TCRs

Thank you for your suggestion. We edited the description in the abstract to “enriched for predicted autoantigen-specificity”.

3. Line 48: Can the authors explain how RFUs differ from a bag of kmers? It would be helpful to have a schematic of the RFU extraction pipeline (e.g. similar to that in reference 18), as this is an important concept for the rest of the article

Although a bag of kmers could be used to provide a numeric encoding for a TCR repertoire, it only provides the frequency of trimers without any additional information. The essence of RFU method is its reliance on a trimer-embedding space derived from mining large-scale TCR repertoire sequencing data. This embedding is built on the trimer-replace matrix, which

represents the frequencies that one amino acid trimer to be replaced by another in the context of shared antigen specificity. Therefore, each RFU carries meaningful information about antigen specificity, unlike a bag of kmers. As mentioned in our response to the major comment, we included a description of RFU in the revised manuscript and a schematic of the RFU pipeline in Supplementary Fig. 1.

4. Line 67: Does aggregating across time points skew the dataset towards patients who are sampled at multiple time points?

We thank the reviewer for raising the concern. After aggregating TCR sequences for each patient, one patient contributed only one data point to the regression. Besides, we filtered the dataset to only retain patients with more than 2,500 TCR sequences, and the PEER factor used to correct RFU abundance was correlated with the TCR number (Supplementary Fig. 3d). Therefore, the number of time points did not skew the dataset.

5. Line 72-4: to what extent does normalisation and PEER adjustment improve performance of the regressor?

Thank you for the question. Normalization and PEER adjustment were used to remove suspicious associations originating from extreme values and confounders. They did not increase the number of significant associations: we identified 4,397 associations without normalization, 6,128 associations without PEER adjustment, and 2,083 associations with normalization and PEER adjustment. Extreme values existed before normalization, potentially driving some associations. For example, rs17433854 was associated with raw RFU 1421 (Fig. R5a, $p = 5.7 \times 10^{-13}$) but not with normalized RFU (Fig. R5b, $p = 0.74$). Besides, PEER factors corrected hidden covariates including the number of TCR sequences (Supplementary Fig. 3d). Thus, normalization and PEER adjustment were necessary for the regressor. We edited the manuscript to make the rationale clear:

“This normalization was necessary to prevent suspicious associations caused by extreme values. Additionally, similar to eQTL analysis, we adjusted for the first two probabilistic estimation of expression residual (PEER) factors to account for hidden confounders, including sequencing depth, to avoid spurious associations (Supplementary Fig. 3d, e).”

Fig. R5 | The associations between rs17433854 and RFU 1421. The top panel displays the RFU motif, while the bottom part illustrates the unnormalized RFU (a) and normalized RFU (b) abundances corresponding to different variants.

6. Line 75: Is the correlation between TRB and RFU not exactly what one would expect, given encoding of TCR germline sequences that will be reflected in the N terminus of the CDR3 (line 80)? What is the novelty of this, and what happens to the significance of non-TCR coding genes when you mask the TRB locus?

We agree with the reviewer that it is known that the DNA variants in the TRB region can be important in regulating V(D)J recombination, and hence will influence the compositions of the immune repertoire. Our association analysis provides a quantitative measurement of how exactly each variant might influence which TCR cluster in the repertoire, which has not been performed before. In addition, although the CDR3 region of TRB genes was reflected in the N terminus of the CDR3, other regions were not. The missense variant we mentioned was in the first framework region, a non-coding region far from the CDR3 region. Thus, it was not as straightforward as reflecting the variant in the CDR3 sequences and was likely to be due to regulatory effects. Since we assessed the associations for each variant separately, non-TCR coding genes did not interact with variants in the TRB locus in our analysis. Thus, the significance would remain the same when masking the TRB locus.

We edited the text to make the novelty clear:

“The most significant association was a missense variant in the non-coding region of TRBV 12-5 associated with the abundance of RFU 1415 (Fig. 1c). Consistently, TCRs assigned to RFU 1415 contained the initial sequence of the TRBV 12-5 CDR3 region (“CASGL”)⁸, suggesting that the variants may regulate the expression of TRBV 12-5.”

7. Fig.1d a smaller dot size on the scatterplot and jitter might make this plot easier to interpret (as for other figures)

Thank you for the suggestion. We tried a smaller dot size but found it did not help with visualization. Since jitter is not commonly used in Manhattan plots, we did not include jitter in the figure.

8. Line 185: The assertion that TCRs assigned to the same RFU will have a 90% chance of having the same antigen specificity is not based on strong evidence, as set out above, particularly given the dataset in question is pre-processed to remove TCRs having the same sequence but different antigen specificity

Thank you for pointing out this concern. As mentioned in our response to your major concern, the assertion was based on multiple randomly selected epitopes. We removed TCRs with the same sequence but different antigen specificity to avoid bias in our evaluation.

9. Line 187: Were there any specific findings for Antigen-specific TCRs drawn from VDJdb (mentioned in the methods)?

Thank you for your question. We did not use antigen-specific TCRs from VDJdb for celiac disease (CD) and type 1 diabetes (T1D) analysis. The numbers of TCR sequences specific to CD and T1D were limited in VDJdb compared to McPAS-TCR. There were only 23 CD-specific and 4 T1D-specific TCRs in VDJdb, while McPAS-TCR had 558 CD-specific and 615 T1D-specific TCRs. The number of TCR sequences specific to autoantigens in VDJdb was not enough for our analysis. We included the explanation in Methods:

“Since the numbers of TCR sequences specific to CD and T1D were limited in VDJdb compared to McPAS-TCR, we only used McPAS-TCR for CD and T1D specific TCRs.”

10. Line 328-9: What was the proportion of exact matches between the training and test sets, and how was clonality determined?

Thank you for your question. 14% of sequences in the test set overlapped with the training set. Since the training and test sets are used to assess the associations between genetic variants and the TCR repertoire, it is necessary to include all TCR sequences in the repertoire, and some exact matches between the training and test sets are expected. Clonality was determined from the processed files of each dataset: the “seq_reads” column for rfuQTL test set, and the “max_productive_frequency” column for ImmuneAccess files. We added the definitions of clonality in Methods:

“For each individual, we calculated the RFU abundance from the 20,000 TCR sequences exhibiting the highest clonality (“seq_reads” column in the processed files) in the TCR-seq data. ...

RFUs were calculated from CD TCR-seq and T1D TCR-seq datasets using the first 10,000 abundant TCR sequences (the “max_productive_frequency” column in ImmuneAccess files), which is the common practice for TCR-seq data analysis.”

11. Line 353-4: How did you handle cases (if any) in which more than one antigen-specific TCR mapped to a given RFU?

We assigned the RFU as specific to all antigens if more than one antigen-specific TCR mapped to it. Since many TCR sequences reacted to multiple antigens due to their known cross-reactivity⁹, it was normal to have more than one antigen specificity for a given RFU.

We added the explanation in Methods: Antigen-specific RFUs.

12. Line 402-405: The methodology for feature selection and train/test validation is not clear. Please provide a concise description of the feature vector that is provided to the model, how and why a lasso model was selected, performance of an appropriate comparator model, and results when there is no overlap between the train and test sets.

Apologies for the confusion in the methodology section. We trained a lasso model for each RFU. The input of the model is the genotypes of all subjects, and the output is the RFU abundance of all subjects. Since lasso models and Elastic Net consistently achieve strong performance in similar tasks, we began with these two models and found that the lasso model performed better. Besides, using heritability estimation as an upper bound for our performance, we found that the lasso model captured most of the variance that could be predicted by genetic variants. Thus, we chose the lasso model.

As mentioned in our response to comment 10, the training and test sets are used to assess the associations between genetic variants and TCR repertoire, so it is necessary to include all TCR sequences. It did not make sense to remove overlapping peptides from the test set.

We edited our methods to clarify the model:

"In rfuQTL training set, we developed both lasso and elastic net models for each variant set and each RFU to predict normalized and covariates-corrected RFU abundances. The input of the model is the genotypes of all subjects, and the output is the RFU abundance for them. Since lasso models and Elastic Net consistently achieve strong performance in similar tasks, we began with these two models and chose the lasso model since it performed better."

Line 414: Why was a different threshold used for the TCR-seq dataset than McPas-TCR/VDJdb?

Thanks for the question. When analyzing TCR-seq datasets, we aimed to obtain the RFU abundances for each sample. We used top 10,000 TCR sequences for RFU assignment to eliminate the influence of sequence depth. When analyzing McPAS-TCR/VDJdb datasets, we aimed to assign RFU for each TCR sequences to obtain the antigen-specific TCRs. We did not need to filter the dataset since we were not trying to calculate RFU abundance for each sample.

References:

- 1 Ishigaki, K. *et al.* HLA autoimmune risk alleles restrict the hypervariable region of T cell receptors. *Nature Genetics* **54**, 393-402 (2022). <https://doi.org/10.1038/s41588-022->

01032-z

- 2 Yu, X. *et al.* Quantifiable TCR repertoire changes in pre-diagnostic blood specimens among high-grade ovarian cancer patients. *bioRxiv*, 2023.2010.2012.562056 (2023). <https://doi.org:10.1101/2023.10.12.562056>
- 3 Bycroft, C. *et al.* The UK Biobank resource with deep phenotyping and genomic data. *Nature* **562**, 203-209 (2018). <https://doi.org:10.1038/s41586-018-0579-z>
- 4 Karczewski, K. J. *et al.* Pan-UK Biobank GWAS improves discovery, analysis of genetic architecture, and resolution into ancestry-enriched effects. *medRxiv*, 2024.2003.2013.24303864 (2024). <https://doi.org:10.1101/2024.03.13.24303864>
- 5 Zhu, C. *et al.* Kidney injury in response to crystallization of calcium oxalate leads to rearrangement of the intrarenal T cell receptor delta immune repertoire. *Journal of Translational Medicine* **17**, 278 (2019). <https://doi.org:10.1186/s12967-019-2022-0>
- 6 Russell, M. L. *et al.* Combining genotypes and T cell receptor distributions to infer genetic loci determining V(D)J recombination probabilities. *eLife* **11**, e73475 (2022). <https://doi.org:10.7554/eLife.73475>
- 7 Yu, X. *et al.* Quantifiable TCR repertoire changes in prediagnostic blood specimens among patients with high-grade ovarian cancer. *Cell Reports Medicine* **5** (2024). <https://doi.org:10.1016/j.xcrm.2024.101612>
- 8 Manso, T. *et al.* IMGT® databases, related tools and web resources through three main axes of research and development. *Nucleic Acids Research* **50**, D1262-D1272 (2021). <https://doi.org:10.1093/nar/gkab1136>
- 9 Nelson, Ryan W. *et al.* T Cell Receptor Cross-Reactivity between Similar Foreign and Self Peptides Influences Naive Cell Population Size and Autoimmunity. *Immunity* **42**, 95-107 (2015). <https://doi.org:https://doi.org/10.1016/j.immuni.2014.12.022>

REVIEWERS' COMMENTS:

Reviewer #1 (Remarks to the Author):

I appreciate the efforts the authors have made to address my previous comments, and I do not have any further concerns.

Reviewer #2 (Remarks to the Author):

Thank you for the comprehensive response, and for taking the time to incorporate appropriate amendments. I am comfortable that my concerns have been duly addressed, and congratulate the authors on the amended manuscript and acceptance of the supporting publication.